# Large-Scale Tungsten Fibre-Reinforced Tungsten and Its Mechanical Properties

Daniel Schwalenberg [1,2,†], Jan Willem Coenen [1,3,*,†], Johann Riesch [4], Till Hoeschen [4,†], Yiran Mao [1], Alexander Lau [1], Hanns Gietl [5], Leonard Raumann [1], Philipp Huber [6,†], Christian Linsmeier [1] and Rudolf Neu [2,3]

1   Forschungszentrum Jülich GmbH, Institut für Energie- und Klimaforschung, 52425 Juelich, Germany
2   Technische Universität München, 85748 Garching, Germany
3   Department of Engineering Physics, University of Wisconsin–Madison, Madison, WI 53706, USA
4   Max-Planck-Institut für Plasmaphysik, 85748 Garching, Germany
5   Fusion Safety Program, Idaho National Laboratory, Idaho Falls, ID 83412, USA
6   Institut für Textiltechnik (ITA), RWTH Aachen University, 52074 Aachen, Germany
*   Correspondence: j.w.coenen@fz-juelich.de; Tel.: +49-246-161-5536
†   These authors contributed equally to this work.

**Abstract:** Tungsten-fibre-reinforced tungsten composites ($W_f/W$) have been in development to overcome the inherent brittleness of tungsten as one of the most promising candidates for the first wall and divertor armour material in a future fusion power plant. As the development of $W_f/W$ continues, the fracture toughness of the composite is one of the main design drivers. In this contribution, the efforts on size upscaling of $W_f/W$ based on Chemical Vapour Deposition (CVD) are shown together with fracture mechanical tests of two different size samples of $W_f/W$ produced by CVD. Three-point bending tests according to American Society for Testing and Materials (ASTM) Norm E399 for brittle materials were used to obtain a first estimation of the toughness. A provisional fracture toughness value of up to 346 MPa m$^{1/2}$ was calculated for the as-fabricated material. As the material does not show a brittle fracture in the as-fabricated state, the J-Integral approach based on the ASTM E1820 was additionally applied. A maximum value of the J-integral of 41 kJ/m$^2$ (134.8 MPa m$^{1/2}$) was determined for the largest samples. Post mortem investigations were employed to detail the active mechanisms and crack propagation.

**Keywords:** composites; fusion; materials

## 1. Introduction

Tungsten (W) components are currently considered as the primary candidate for the first wall and divertor armour of existing and future fusion reactors. Tungsten has a very low sputtering yield and the highest melting point of any metal, and it behaves relatively well in terms of its properties after neutron irradiation. With respect to the interaction with the fusion fuel, tungsten shows low retention of hydrogen isotopes, including tritium.

Much of the existing work regarding tungsten has focussed on the qualification of ITER (https://www.iter.org (accessed on 27 October 2022)) [1–4] and beyond. For the next steps, e.g., a Demonstration Reactor (DEMO), (https://www.euro-fusion.org/programme/demo/, (accessed on 27 October 2022)) the imposed limits of the use of tungsten will be extremely challenging as much of what is determined as boundary conditions [5,6] materials will be above the technical feasibility limits as they are set out today [3,7].

New concepts for plasma-facing components (PFCs) are being studied (see [2,8] and references therein) concentrating on crack-resilient materials with low activation, small or no fuel retention, extended lifetime with respect to low erosion, and brittle failure. This is due to the fact that the inherent brittleness below the ductile-to-brittle transition temperature of tungsten (DBTT) [9,10] and the embrittlement during operation, e.g., by

overheating [11] and/or neutron irradiation [12,13], are the main drawbacks and limit for its use in terms of lifetime considerations.

In Figure 1, the typical properties necessary for fusion are highlighted. The core blue area is the status quo.

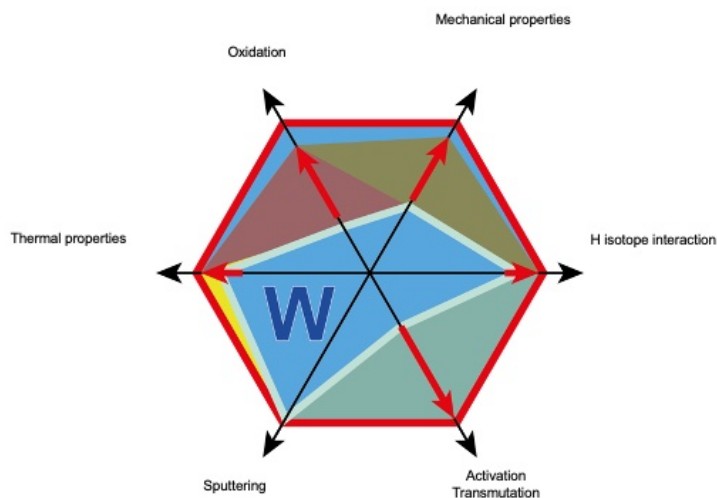

**Figure 1.** W, the first wall and divertor material, still faces challenges. The red lines give the performance with respect to ideal behaviour, represented by the outline of the hexagon. The core blue area is the status quo [8].

In essence, the areas of oxidation and mechanical properties need to be improved upon in particular.

In this contribution, the efforts on size upscaling of $W_f/W$ based on Chemical Vapour Deposition (CVD) are shown followed by the description of fracture mechanical tests of CVD $W_f/W$.

## 2. Tungsten-Fibre-Reinforced Tungsten

To overcome the brittleness of W a composite material $W_f/W$ incorporating extrinsic toughening mechanisms as described in [8,14], is being developed. The basic concept of materials such as $W_f/W$ makes use of a composite approach, as highlighted in Figure 2a.

The issue with tungsten essentially is the same as for all brittle materials. Failure immediately and suddenly occurs when the ultimate tensile strength (UTS) is reached. UTS here is statistically distributed and not well defined, as it is determined by the weakest point in the material (Weibull statistics). For a materials with well-defined material properties, with a defined yield point, rising load bearing capability, and, as such, high toughness and ductility, the material is intrinsically failure-tolerant, with no sudden failure, and it remains load bearing, beyond UTS, also allowing cyclic loading of the material.

In order to improve brittle materials such as W, several options are available: microstructural refinement, alloying, or composites [15–21]. Regarding $W_f/W$, extrinsic toughening mechanisms are employed, thus potentially allowing material changes due to operational embrittlement by temperature and neutrons to be overcome.

When considering $W_f/W$, it is important to note that there are various methods of production including CVD [22,23] and powder metallurgical (PM) processes [24–28]. The chemical vapour deposition of tungsten is frequently used for the production of $W_f/W$ as it allows low processing temperatures and force-free production. The process used is the heterogeneous surface reaction of $WF_6$ and $H_2$ to form a solid W deposit and gaseous HF. This process is highly sensitive to the partial pressures as well as temperatures, and the details are given in [29–31]

Based on existing work [32–38], the basic proof-of-principle for both $W_f/W$ materials (PM and CVD $W_f/W$) was given. The typical structure of the composite is based on wires

coated with an oxide ceramic, e.g., $Y_2O_3$ and a W-matrix. The interface is the main vehicle to enable pseudo-ductile behaviour. $W_f/W$ shows pseudo-ductile behaviour even at room temperature for both the PM and CVD routes. This means that despite crack formation, load bearing capability is retained.

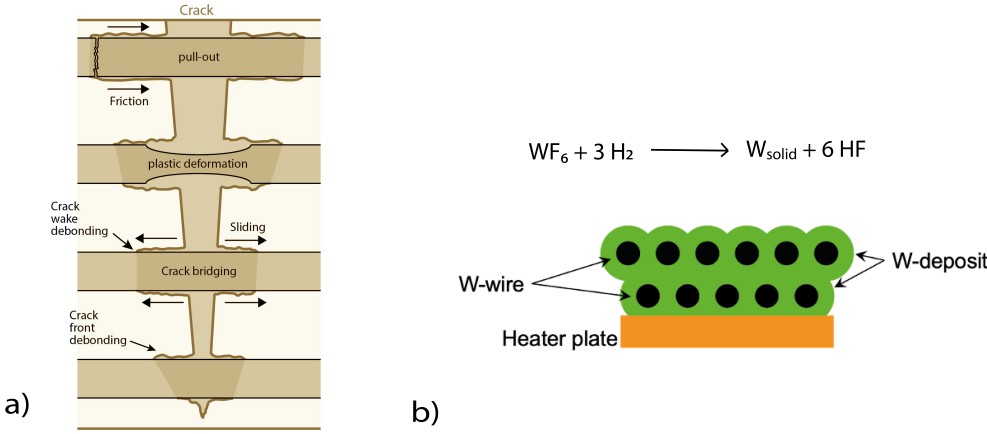

**Figure 2.** (**a**) A selection of energy-dissipation mechanisms in a fibre-composite material. Pull-out of fibres, pull-out of matrix elements, crack deflection at the interface, crack bridging by fibres, crack meandering at the interface, and plastic deformation of fibres (based on [8,36]). (**b**) Process of layer-wise growth of $W_f/W$ by applying one layer of weave at a time.

When optimising the production process for $W_f/W$, it is essential to retain the properties of the constituents (e.g., wires) as much as possible to allow for optimal extrinsic toughening and pseudo-ductility. The interface used as well as the strength of the wire in the preform [33] are important for the overall material properties. Typically, Yttria is used as an interface material for the $W_f/W$ composite due to its several advantageous properties: good thermal and chemical stability, high mechanical strength, and hardness [39,40], and low neutron activation. To improve the CVD $W_f/W$ material, multiple avenues are being pursued, including CVD parameter optimisation [29–31].

In recent times, both PM and CVD processes have been scaled to allow larger sample production. In this contribution, the focus lies on CVD-based $W_f/W$ and its upscaling. In the past [35], the typically available dimensions were limited to $60 \times 60$ mm$^2$. As described below, the samples that are available now reach $190 \times 60$ mm$^2$ and more. For the $W_f/W$ described here, no Yttria interface was applied due to the difficulty of applying it to large preforms. Work on upscaling and optimising the interface coating procedures for Yttria is ongoing.

### 3. Materials and Methods of Sample Production

The $W_f/W$ investigated here was produced in a layer-wise process (Figure 2b) as already described in [14,35,36]. The temperature for the process was approximately 873 K at a pressure of 100 mbar and a coating time per layer of 78 min. The precursor gas was heated to about 523 K to allow for a more homogenous coating of the whole preform. The volume flow of $H_2$ and $WF_6$ was 25,000 sccm and 2000 sccm, respectively.

The weaves (Figure 3) are produced from a 150 μm potassium doped warp wire and a 50 μm potassium weft wire produced by OSRAM GmbH. For the production of dense $W_f/W$, the optimal distance between the fibres for 150 μm is around 350 μm as described in [33]. The preforms are produced at ITA Aachen, where a Mageba shuttle loom (type SL 1/80) (Mageba International GmbH, Bernkastel, Germany) weaving machine was used to produce the weaves shown in Figure 3.

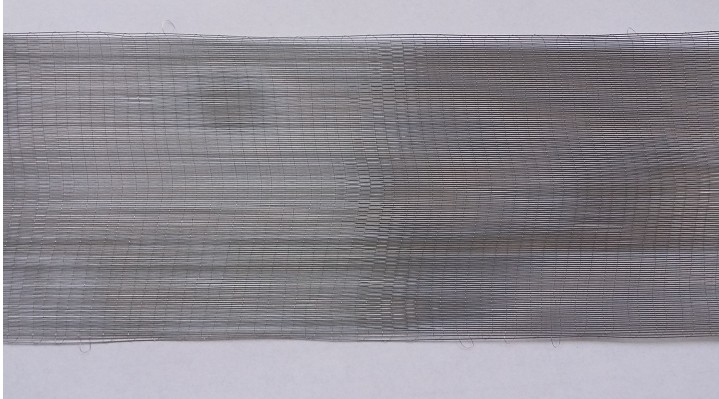

**Figure 3.** Tungsten weaves with 150 μm wires utilised to form a 50 mm wide weave.

In contrast to the samples produced and described in [14,35,36], the heating table employed had a dimension of 150 × 150 mm, employing a 5 mm thick 200 × 200 mm² tungsten plate as substrate holder. Figure 4 shows the process of placing the woven preform in two strands every 50 × 200 mm² on top of the previously coated material.

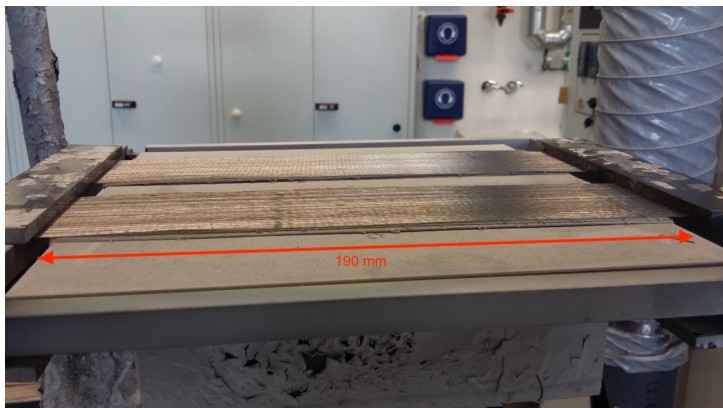

**Figure 4.** Heating table incl. tungsten substrate plate and Frame with weave.

In a similar manner, all 23 layers of weaves are incorporated into the composite to finally form the 10 mm thick composite. During production, it was already seen that one of the composite blocks showed layer delimitation after about 20 layers (Figure 5), a typical complication of the layer-wise process.

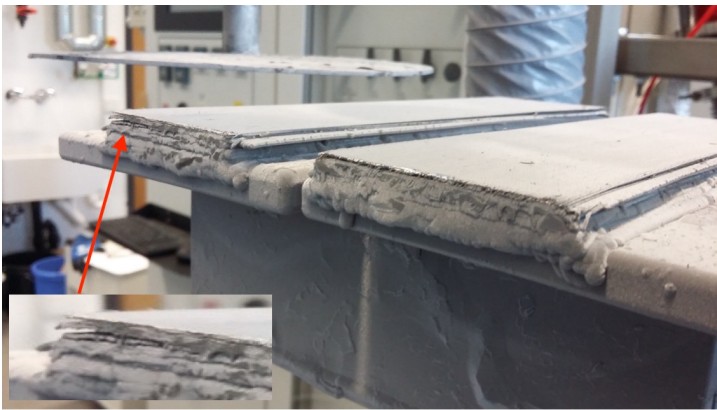

**Figure 5.** Side view of the samples produced after 20 layers.

Due to the varying layer quality, care was taken to cut the samples used later from the highest quality areas. Areas with lower quality were used to cut smaller samples. From

the composite block with good quality, the larger samples discussed here were produced. The right sample was used to cut KLST-type samples comparable to the work in [14]. Here, one difference needs to be mentioned. Whereas in the previous works related to CVD $W_f/W$, the layers of the composite were often parallel to the top surface of the three-point bending samples, this time, they are parallel to the sides. This might not be much of a concern for an ideal material but will have benefits, as we will see later for the large samples used here.

Cutting was performed via electro discharge machining following the plan in Figure 6, and due to delamination issues, only one large and five medium-sized viable sample were produced. A pre-notch was applied using the same process. In Table 1, the dimension and general properties are given.

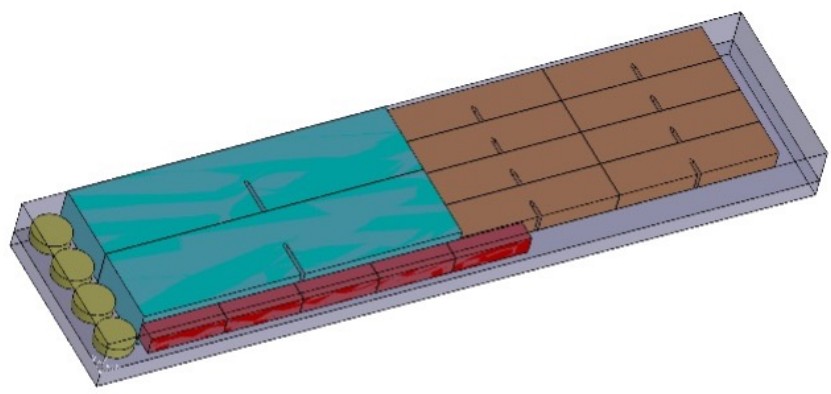

**Figure 6.** Cutting -plan for samples based on quality of layers.

**Table 1.** Samples produced with their repective dimension and properties.

| Sample Type | Dimension (Length × Width × Height) | Amount | Relative Density | Fibre-Volume-Fraction |
|---|---|---|---|---|
| Large | 84 × 24 × 10 | 1 | 92 | 10.4 |
| Medium | 42 × 12 × 5 | 5 | 88–93 | 10.4–12.16 |

In comparison to [14], the samples are significantly larger while retaining similar density and fibre volume fraction.

## 4. Mechanical Tests

In order to compare the large-scale samples in this work with the previous fracture toughness determinations [14] the three-point bending test were performed with the same setup. A TIRAtest 2820, Nr. R050/01 from TIRA GmbH was utilised at room temperature (RT). To determine displacement and crack growth on the surface, the load-displacement curves were correlated to an optical surface observation. For the optical surface observation, an optical measurement system with a tele-centric lens (OPTO ENGINEERING—TC4 M004-C) with a four times magnification was used in combination with a monochrome digital camera (Toshiba—Type DU657M). For  all mechanical tests, a 5 kN load cell was used. Following ASTM E399, the spans for the different size samples were chosen to be 80 and 40 mm, respectively. Following the ASMT 399 norm, all samples were prepared with a pre-notch. Due to the use of EDM machining, the notch tip was rounded with a 1 mm EDM wire. The notch's depth was always 50% of the sample height. The notch width was 1.155 mm. No additional Razor-blade or FIB sharpening was performed, because considering results from previous studies [14], the need for sharpening was not given. A sketch of the sample shape is shown in Figure 7.

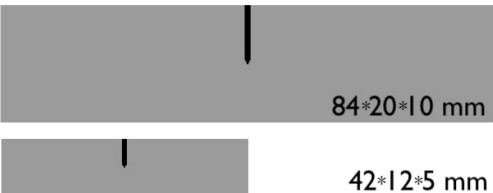

**Figure 7.** Sketch including pre-notch and dimensions for both the large and the medium-sized samples.

The tests were performed with a constant displacement rate of 30 µm for the large samples and 5 µm for the medium samples. The camera operated at 50 Hz.

## 5. Fracture Toughness Determination

Originally, the intention was to only utilise the ASTM E399 norm. However, during the study, it became apparent that the sample size was too small to perfectly fulfil the size requirements of ASTM E399; thus, the data were additionally used to perform a $J_Q$ calculation. This also is helpful due to taking into account the ductile nature of the force-displacement curves.

### 5.1. ASTM E399

The fracture toughness values according to ASTM E399 were calculated by analysis of the load–displacement curves, with the following equation:

$$K_P = \frac{P \cdot S}{B \cdot H^{3/2}} \cdot f(a/H) \tag{1}$$

where $K_P$ is a provisional fracture toughness, $P$ is the load applied to the specimen, $S$ is the span, $B$ is the thickness, $H$ is the height, $a$ is the crack length, and $f(a/H)$ is a dimensionless function defined in the standard [41].

$K_P$ is for this specimen geometry defined as $K_{1C}$ if two size criteria are fulfilled according to [41]. The first one defines the specimen height-to-crack-length ratio ($0.45 \leq a/H \leq 0.55$). The second size criterion defines the crack length a and the specimen thickness B. Both have to be larger than $2.5 \times (K_P/\sigma_y)^2$

For an inherently three-dimensional inhomogeneous material such as $W_f/W$, the determination of the actual crack length (a) becomes complicated as the surface crack does not necessarily reflect the actual crack length [42]. Despite that fact, we utilised the visible crack at maximum force to achieve a guidance on $K_P$ determined as $K_{R_{max}}$. The data used for the ASTM E399 calculation are given in Table 2.

**Table 2.** Data for ASTM E399 calculation.

| Sample Type | B [mm] | H [mm] | $a_{notch}$[mm] | $a_{max}$[mm] | f(a/W) | $F_{max}$[N] | $K_{R_{max}}$[MPam$^{1/2}$] |
|---|---|---|---|---|---|---|---|
| Large Sample | 84 | 10 | 10 | 8 | 30.83 | 4333.53 | 346.68 |
| Medium Sample 1 | 42 | 12 | 5 | 3.5 | 16.69 | 1218.38 | 162.68 |
| Medium Sample 2 | 42 | 12 | 5 | 3.5 | 16.69 | 939.03 | 125.38 |
| Medium Sample 3 | 42 | 12 | 5 | 3.75 | 22.03 | 1218.93 | 214.78 |
| Medium Sample 4 | 42 | 12 | 5 | 3.5 | 16.69 | 790.86 | 105.6 |
| Medium Sample 5 | 42 | 12 | 5 | 3.5 | 16.69 | 806.25 | 107.65 |

### 5.2. ASTM 1820

As stated in [43], and reiterated in [14], the classical application of the J-Integral is a path-independent value of the stress concentration around, but excluding the crack tip. However, this only holds true with respect to path independence for straight cracks in homogenous materials with unloaded crack surfaces [44]. However, one can calculate a global J-Integral that is also applicable for composites [45] by accounting for the global

amount of energy absorbed by the specimen. Here, the loading of the crack surface has to be taken into account as well. This is due to the fact that the energy absorbed within a composite material is not only in the crack tip but also behind the matrix crack tip, where extrinsic toughening is at play, such as delamination, pull-out, fibre deformation and fibre fracture.

The $J_Q$ values given in this manuscripts are based on ASTM E1820 and [14], with the caveat that the specimens were not cyclically loaded and unloaded, to calculate $J_Q$, but the already-utilised data of the three-point bending tests described above was employed. This provides guidance for the values for the two applied norms.

The J-Integral is calculated as an elastic $J_{el}$ and a plastic $J_{pl}$ part as follows [46]:

$$J_{(i)} = J_{el(i)} + J_{pl(i)} \tag{2}$$

$$J_{el(i)} = \frac{(K_{(i)})^2 \cdot (1 - \nu^2)}{E} \tag{3}$$

$$J_{pl(i)} = [J_{pl(i-1)} + \frac{\eta_{pl(i-1)}}{b_{(i-1)}} \cdot \frac{A_{pl(i)} - A_{pl(i-1)}}{B}] \times [1 - \gamma_{pl(i-1)} \cdot \frac{a_{(i)} - a_{(i-1)}}{b_{(i-1)}}] \tag{4}$$

Following work in [14] as well as [46], one can calculate a stress intensity factor utilising $J_{(i)}$ in the following names $J_Q$

$$K_Q = \sqrt{\frac{J_Q \cdot E}{1 - \nu^2}} \tag{5}$$

with a valid test only given when fulfilling the size criterion $B > 10\frac{J_Q}{\sigma_y}$

$K_J$ is calculated with the respective span given above. $\eta_{pl}$ and $\gamma_{pl}$ are related to crack length and original specimen dimension whereas $A_{pl}$ is the area under the load displacement after the elastic part is removed. E is typically the Young's modulus with $\nu$ as the Poisson ratio. $i$ in this case gives the steps of the crack propagation utilised in the calculation instead of the cycle number. The final crack length is determined only by the surface crack length. The Young's Modulus used is 405 $GPa$ with the Poisson ratio being 0.3 based on [10]. The data used for the ASTM 1820 calculation are given in Table 3.

**Table 3.** Data for ASTM1820 calculation.

| Sample Type | B mm | H (mm) | $a_{notch}$ (mm) | $a_{max}$ [(mm)] | $J_{Qmax}$ [KJ/m²] | $K_{Qmax}$ [MPam^{1/2}] |
|---|---|---|---|---|---|---|
| Large | 84 | 10 | 10 | 8 | 40.84 | 134.82 |
| Medium 1 | 42 | 12 | 5 | 3.5 | 15.11 | 82.01 |
| Medium 2 | 42 | 12 | 5 | 3.5 | 8.81 | 62.61 |
| Medium 4 | 42 | 12 | 5 | 3.5 | 8.36 | 61.01 |
| Medium 5 | 42 | 12 | 5 | 3.5 | 6.8 | 55.01 |

## 6. Experimental Results and Discussion

To complement the calculations given above in the following, the actual force displacement curves, as well as the pictures of the samples at various steps of crack opening, are given together with the $J_{pl}$ curves.

### 6.1. Large Sized Sample

For the large sample, it is quite apparent that despite dealing with a composite material, the evolution of the force-displacement curve as given in Figure 8 appears very smooth, which is indicative of the large sample size, whereas the individual elements containing fibres and matrix contributed to the overall pseudo-ductile behaviour in an incremental way.

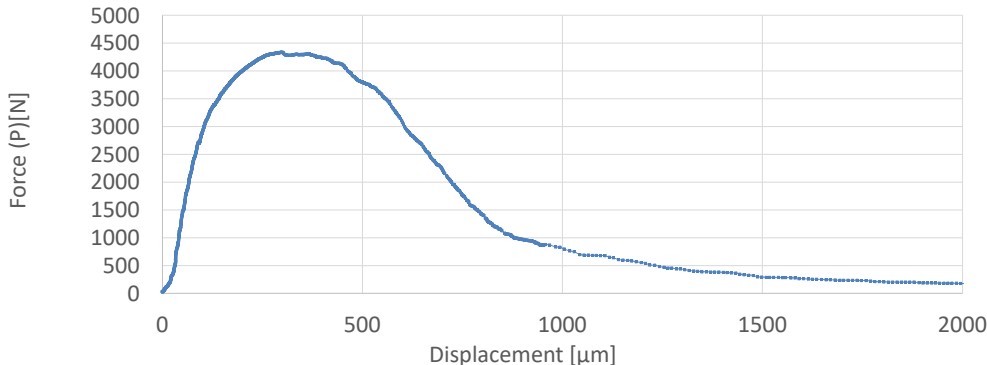

**Figure 8.** Force–displacement curve of large sample.

Looking at the details of the fracture surfaces as given in Figure 9, one can see a few important features. For example, on the right-hand side, the initial notch is visible. The fracture surface is largely smooth; however, at the top of the picture, some of the fibre elements are not level with the rest of the fracture surface. After identifying which fibres showed necking and which did not, it can be said that 82% of the fibres show necking, which is in good agreement with the pseudo-ductile behaviour of the force-displacement curve.

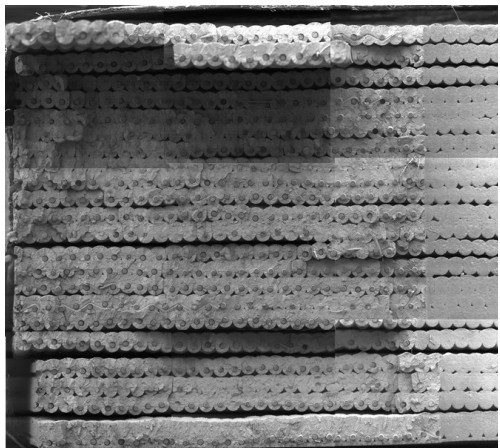

**Figure 9.** Composite picture of the fracture surface of the large sample with FVF and density as given in Table 1.

What is also visible is that—as expected—in this large sample, not all layers are ideally connected. Here, improvements are needed both in processing as well as textile preform manufacture to overcome this. First steps to this end are studied with respect to the use of yarn-based performance as described in [47,48].

In contrast to previous studies such as [14], it is also visible that the individual layers are parallel to the side surfaces of the samples instead of the top. Here, this mitigates the issue of interlayer delamination and potentially improves the material properties.

In Figure 10, the crack opening of the large sample is shown. As already seen in the force-displacement curves, the material shows stable crack growth almost all the way through the sample. The pictures shown are cropped to cover the best view of the opening crack.

Using the data obtained from Figure 10 and the sample properties, the plastic part of the J-Integral can be calculated as seen above the values are shown in Figure 11.

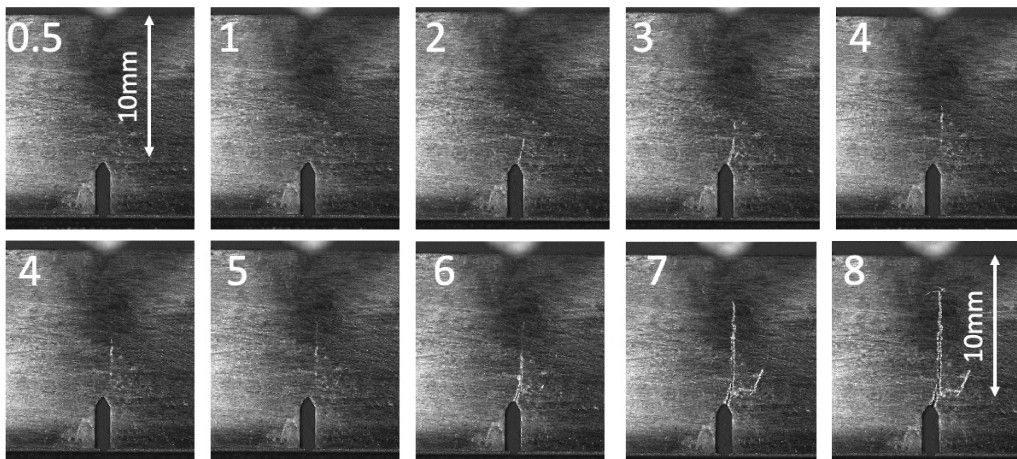

**Figure 10.** Crack propagation during a 3-point bending test of a large sample, with numbers indicating the crack opening measured from the pre-notch.

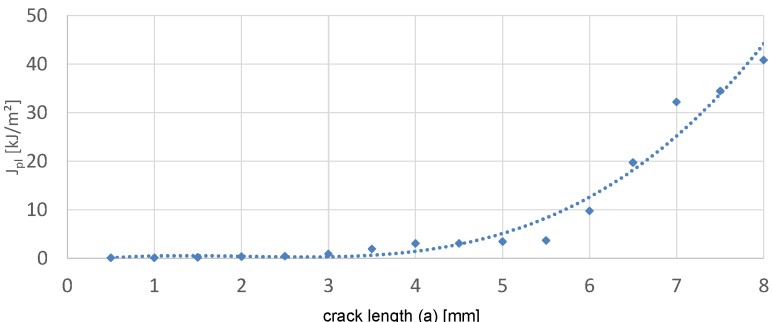

**Figure 11.** $J_{pl}$ for the large sample calculated based on the sample parameters and the crack opening shown in Figure 10.

The evaluated $K_R$ (Figure 12) curves and the $J_{pl}$ curves (Figure 11) for the large sample basically have a concave shape with some small plateaus. This is a mixture of the behaviour seen in [14], where no plateaus were seen, and the expectations for R curves for homogeneous ductile materials, which show a convex shape with a plateau at larger crack lengths [49].

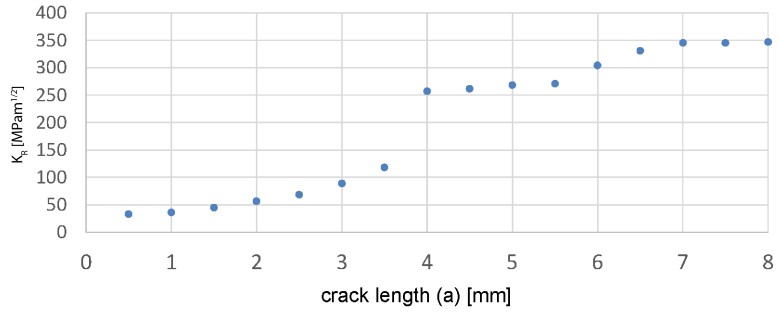

**Figure 12.** $K_R$ based on ASTM E399- values for each step of the crack opening calculated following the large samples as shown in Figure 10.

This indicates that both behaviours are present here. Due to the large sample size and number of unit cells (ductile fibres with matrix), resistance against crack growth is similar to that with homogeneous ductile materials, where the resistance reaches a maximum value for a defined crack length and is overlapping with the behaviour for fibre reinforced composite where for $W_f/W$ the extrinsic toughening mechanisms become more and more

active behind the crack tip, and therefore, the resistance against crack growth rises as the crack grows. This behaviour is caused by the large-scale bridging conditions and is known from other fibre-reinforced composites [50]. In particular,the crack bridging and ductile deformation of the fibres which become active at large crack openings are the major contribution to the toughening.

### 6.2. Medium-Sized Samples

The variability in the behaviour of the medium-sized samples is quite apparent. In Figure 13, the different force–displacement curves show what is already known about the material. Samples 1 and 3 are fairly similar to the large sample as they are seemingly produced with relatively homogenous layer structures and thus show a very smooth trend.

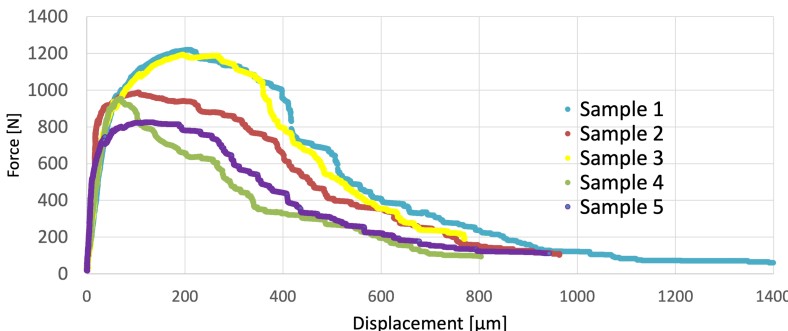

**Figure 13.** Force-displacement curves of medium sized samples.

Due to the limited sample sizes, one can see, however, larger drops in force due to the loss of fibres or fibre bundles in the composite, a typical behaviour for pseudo-ductile materials. The maximum forces of both samples are in the range of 1200N. For the the other samples, either a lower maximum forces (Samples 2, 5) or a much lower load drop are visible.

This can be explained by studying the individual crack surfaces shown in Figure 14.

Where samples 1 and 3 show few pores and a high portion of ductile failure of the fibres leading to a high level of strength, the other samples show much lower quality bonds between the layers, as well as lower density. For samples 2, 4, and 5, one can thus see that the redistribution of force leads to local maxima and more steep drops during the 3-pt bending test.

With respect to the fracture mechanical properties, for the larger samples, more or less stable crack growth is observed, as shown for sample 1 in Figure 15.

The opening of the crack is more difficult to follow, and thus only one example is given here, where the growth is most strongly visible. Based on these pictures, a K value is calculated for the medium-sized samples and plotted against the crack progress. Sample 3 is excluded as the optical data are not sufficient.

Similarly to the case of the large-scale sample, Figure 16 displays the increase in the K value with increasing crack opening.

In contrast to the large sample, no plateaus are visible. It seems, thus, that here, only the extrinsic toughening mechanisms are the main driver for the pseudo-ductile behaviour, while for the large samples, the fibres are even more important in terms of material toughening.

For the medium-sized samples, the $K_{Rmax}$ values range between 105 and 162 MPam$^{1/2}$.

From the $J_{pl}$ values as depicted in Figure 17, a similar assessment compared to the large samples can be made. The drivers of the toughening in both cases are the extrinsic mechanisms supplied by the composite in front of the crack tip.

The likely reason for the strong variations between the various curves is due to the quality of the material that still needs to be improved, as also visible in Figure 15.

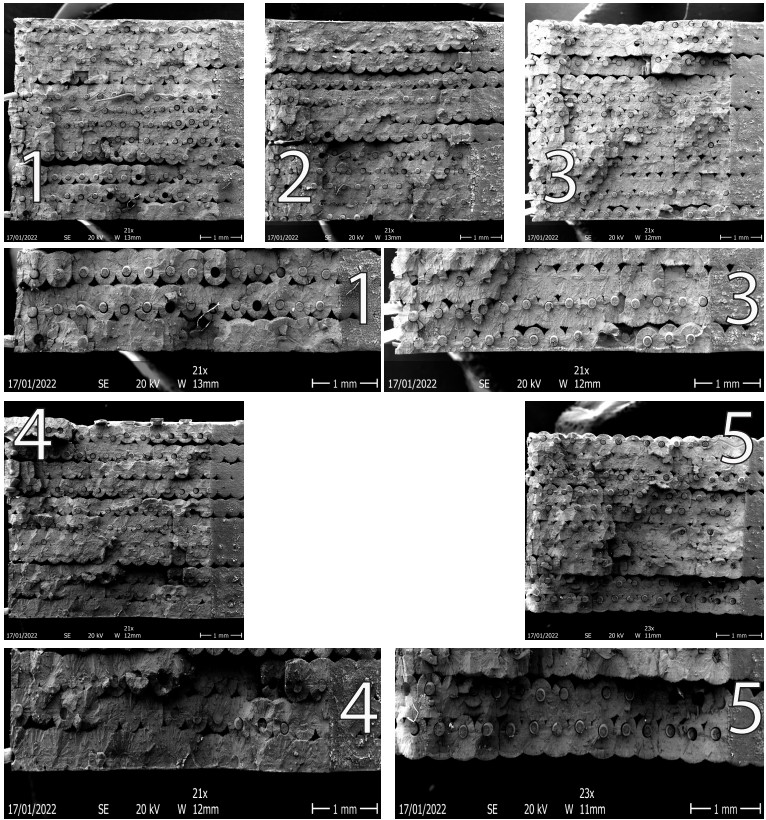

**Figure 14.** Pictures of the fracture surface of the individual medium-sized samples (numbers indicated sample name), the closeups highlight the good ductile failure of the fibres in some cases and the brittle nature in others.

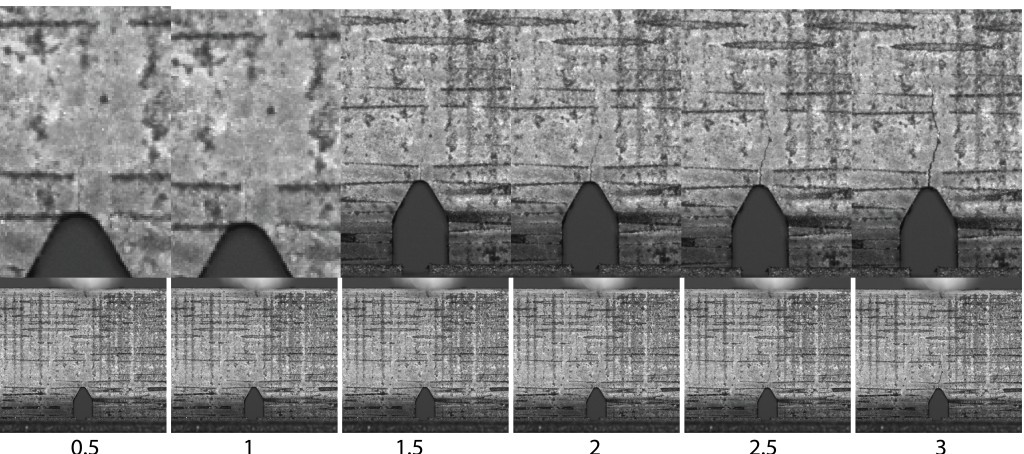

**Figure 15.** Crack opening for Sample 1, including closeups of the crack regions. Resolution is limited due to the camera used. (numbers indicate crack length

The $J_{Q_{max}}$ values range between 6.8 and 15.11 kJ/m$^2$ with $K_Q$ ranging between 55 and 82 MPm$^{1/2}$.

As a rule, it seems that for $W_f/W$, the extrinsic toughening mechanisms such as delamination, pull-out, fibre-deformation, and fibre-fracture all contribute. The ductility of the fibre is one of the main mechanisms and becomes more important the larger the sample becomes.

In all cases, it seems that crack deflection, as well as ductile deformation of fibres, is a main contributor, as also shown in previous work [14].

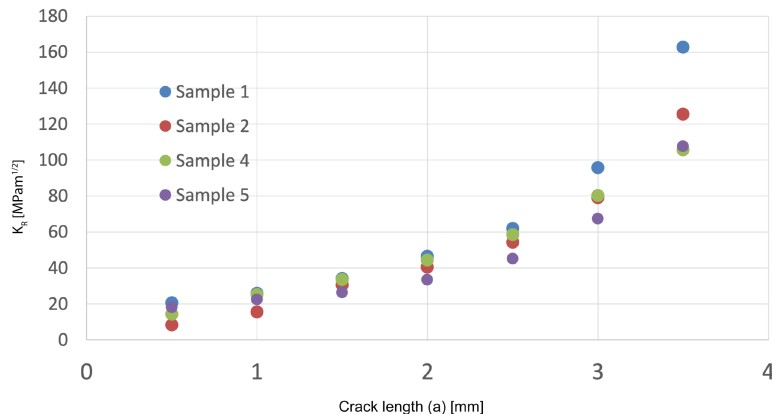

**Figure 16.** $K_R$ values for each step of the crack opening calculated based on ASTM E399 following the medium-sized samples. Sample 3 is missing as the crack opening could not be followed.

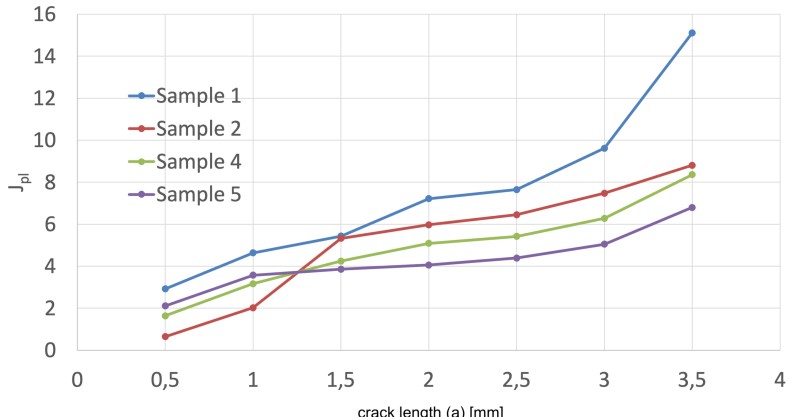

**Figure 17.** $J_{pl}$ for the medium-sized samples calculated based on the sample parameters and the crack opening similar to Figure 15.

Figure 18 shows a close up of the fracture surface of one of the medium-sized samples, which clearly highlights the ductile nature and contribution of the fibres.

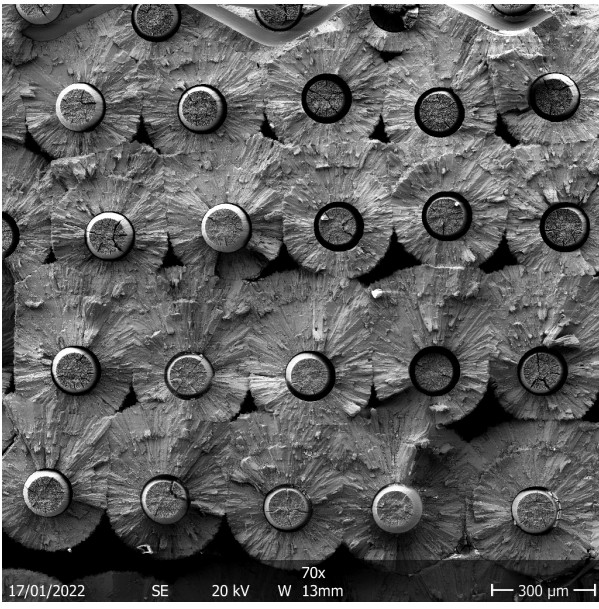

**Figure 18.** Fibre necking visible in the crack surface of the medium-sized sample 1.

## 7. Summary and Conclusions

The aim of this work was to investigate the fracture behaviour and to achieve a first estimation for the fracture toughness of large-scale long-fibre $W_f/W$, complementing the work in [14]. According to the ASTM E399, a maximum value for the large sample of $K_{R_{max}} = 346$ MPam$^{1/2}$ was calculated. As the material showed a stable crack prorogation, the J-Integral approach according to ASTM E1820 was additionally applied to both sample types and a maximum value from the large sample of $J_{Q_{max}} = 41$ kJ/m$^2$ (134.8 MPa m$^{1/2}$) was determined. Even though these samples are significantly larger than in previous studies, the values given here need to be treated with care. Size criteria to fulfil the norms were not met. From the fracture surfaces, it is obvious that for the materials tested here, the main contribution for the pseudo-ductile behaviour originates from the fibre properties and increases the more the crack opens. This is similar to work in [14]. The composite showed stable crack propagation, crack deflection, and crack stopping for all tested samples. It needs to be iterated that in the case of the material studied here, no interface was used. Nevertheless, pseudo-ductile behaviour is observed.

To date, no criterion was identified that describes whether the main contribution is due to the extrinsic toughening mechanisms or the ductile fibre behaviour.

From the presented study, a first estimate for actual component design can be taken as the sample sizes are similar to the typical dimensions of tungsten components in ITER [51] and DEMO [52].

**Author Contributions:** D.S. and J.W.C.: methodology, investigation, visualization, writing-original draft; J.R. and L.R.: conceptualization, supervision; Y.M.: conceptualization, methodology; H.G.: methodology, conceptualization; T.H.: methodology, investigation, visualization, P.H.: methodology, investigation; A.L.: methodology, investigation C.L.: funding acquisition, project administration; R.N.: PhD supervision. All authors contributed additionally with scientific discussions, ongoing feedback and suggestions, and with writing—review and editing. All authors have read and agreed to the published version of the manuscript.

**Funding:** This work has been carried out within the framework of the EUROfusion Consortium, funded by the European Union via the Euratom Research and Training Programme (Grant Agreement No 101052200 EUROfusion). Views and opinions expressed are, however, those of the author(s) only and do not necessarily reflect those of the European Union or the European Commission. Neither the European Union nor the European Commission can be held responsible for them.

**Conflicts of Interest:** The authors declare no conflict of interest.

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
