# Peer review of "Large-Scale Tungsten Fibre-Reinforced Tungsten and Its Mechanical Properties"

_jne, doi:10.3390/jne3040018_

Round 1

Reviewer 1 Report

This manuscript studied the fracture properties and crack behavior of size upscaled Wf /W composites. Three-point bending tests were conducted abiding ASTM E399, and J-Integral method stipulated in ASTM E1820 was adopted to quantify pseudo-ductile of as-fabricated material. The paper has good engineering significance by contributing a first estimation on the fracture toughness of large scale long fiber Wf /W. There are some issues to be justified before considering publication in Journal of Nuclear Engineering.

1.     Comment: Line 127, it is good to know the reason of not performing Razor-blade or FIB, but the discussion on test scenarios should better be kept between authors instead appearing in a written version.

2.     Comment: Line 133, this sentence seems to be wrongly edited, which makes it hard for this reviewer to understand, please check.

3.     Comment: Line 172, please check the value of Young’s Modulus. References on the specific value of Poisson’s ratio are also needed.

4.     Comment: Figure 12, the unit of the y-coordinate is missing, same problem in Figure 17.

5.     Comment: Line 227, it is really hard to judge the relative size or density of pores of all 5 fractures, and the portion of ductile failure requires further instructions. The authors may also have noticed that one major difference between sample 1&3 and others is the more compact structure at the bottom of prefabricated notch (right side of per SEM photograph).

6.     Comment: Line 241, spelling error, “tougnening” would be “toughening”. Please check the manuscript carefully.

7.     Comment: Figure 16, this reviewer understands how hard it is to collect high-quality pictures of the continuous failure process of composites during an in-situ tensile test. However, the resolution of present pictures is insufficient for readers to distinguish the early growth of cracks. Adjust the brightness/contrast of pictures or add necessary annotations could help.

8.     Comment: “Summary and conclusion”, the present study reveals that the fiber properties is responsible for the pseudo-ductile behavior of large scale Wf /W, and the pseudo-ductile of medium size sample is purely driven by the extrinsic toughening mechanisms. Is there any characteristic feature or parameter capable of describing the transition of mechanisms? It seems the contribution of fiber properties on the pseudo-ductile is becoming more outstanding with the propagation of cracks, but the extrinsic toughening mechanisms were further removed from importance.

Author Response

Please find the rebuttal in the attached document

Reviewer 2 Report

the authors show CVD grown of large-scale tungsten fibre-reinforced tungsten and its mechanical properties in this work. The manuscript can be accepted after the following questions are concerned.

1.       The full name should be written when the abbreviation first appears, such as in abstract “The ASTM E399”, the introduction part “ITER”……

2.       Figure 1 seems not necessary; it is just like a Scientific knowledge and should be added in the supporting information.

3.        From the perspective of aesthetics, Figures 2 and 3 should be integrated into one figure, rather than two separate figures. Other figures are also suggested to be re-designed; Figure 11 should be re-designed and they are too small.

4.       There are many formatting errors in the paper, including missing spaces between numbers and units, such as line 84, 93, 95…  Figure captions in Figure 4 and many other parts.

5.       The chemical reaction equation in Figure 3 is incorrectly formatted, such as subscripts and spaces

6.       In Figure 12, there is a In Figure 12, there is a mutation between crack length of 3 to 4, can the author explain why?

7.       In Figure 18, why are the trends very different between parallel samples?

Author Response

(The authors gave the same response as above.)

Round 2

Reviewer 2 Report

It should be published